# Global increase in rain rate of tropical cyclones prior to landfall

Quanjia Zhong [1], Jianping Gan [1,2] ✉, Shifei Tu [3], Ralf Toumi [4] &
Johnny C. L. Chan [5,6,7] ✉

Most studies on tropical cyclone (TC) rain rate focus on long-term variability, yet the short-term (days or shorter) variations across the TC lifecycle, with a particular focus on the period before landfall, are most critical because they strongly influence flood risk. Using satellite data, we show that, globally, the mean rain rate of TCs increases by over 20% from 60 hours before landfall to the time of landfall. This increase occurs across hemispheres, ocean basins, intensity categories, and latitudes, although the magnitude varies. As a TC approaches the coast, land-sea thermal contrasts raise low-level humidity over land, while frictional differences enhance convergence, upward motion, and instability on the offshore side of the circulation. These conditions collectively promote increased convection and precipitation of TCs as they near landfall. Our findings critically strengthen the current understanding of TC precipitation dynamics and support more effective flood management.

Heavy rainfall from landfalling tropical cyclones (TCs) often produces urban waterlogging, flash floods, and landslides; all pose significant threats to coastal communities worldwide[1–3]. When the wind fields and outer rainbands of a TC first reach land, coastal communities feel the effects of the TC even before it makes landfall[4]. The rain rate of a landfalling TC is one of key factors that determines the accumulated precipitation. In addition, when the TC stalls or passes over a given region, during the time it takes for the TC to pass, the accumulated precipitation increases. Because effective flood management and adaptation strategies must be prepared well in advance, it is essential that the position and precipitation of a TC are accurately predicted. Therefore, to accurately predict the precipitation, it is even more essential to know how the rain rate changes as the TC approaches the coastline.

Most studies have revealed that, in a warmer environment, global TCs move more slowly[5–7], and their annual-mean rain rates tend to increase[8–10]. The stalling of a TC over coastal regions, combined with higher rain rates, leads to prolonged heavy rainfall and an increased

risk of flooding[11]. More recently, several studies have investigated how TC rainfall climatology varies across different surface types, such as over the ocean, near land, and over land[12,13]. Collectively, these findings have advanced our understanding of how TC rainfall responds to long-term climate change[14], and how mean rain rates vary with surface conditions.

By contrast, short-term changes in rain rates during the lifecycle of individual TCs, especially at hourly to daily timescales as storms approach land, have received comparatively little attention[13,15]. Yet these transient variations are more directly relevant to real-time emergency response and flood preparedness. Moreover, the physical mechanisms driving these short-term changes remain poorly understood. Here we show through both observational analyses and numerical simulations that TC rain rate prior to landfall increases by about 20% globally, which can be explained based on the changes in the relative humidity, convergence, vertical velocity at low-levels, and instability in this coupled land-sea system. Our findings advance our understanding of the dynamics of landfalling TCs and provide a

¹Center for Ocean Research in Hong Kong and Macau (CORE) and Department of Ocean Science, Hong Kong University of Science and Technology, Hong Kong, China. ²Department of Mathematics, Hong Kong University of Science and Technology, Hong Kong, China. ³South China Sea Institute of Marine Meteorology/Western Guangdong Key Laboratory of Marine Meteorological Disaster Theory and Application, College of Ocean and Meteorology, Guangdong Ocean University, Zhanjiang, China. ⁴Department of Physics, Imperial College London, London, UK. ⁵Asia-Pacific Typhoon Collaborative Research Center, Shanghai, China. ⁶Shanghai Typhoon Institute, China Meteorological Administration, Shanghai, China. ⁷School of Energy and Environment, City University of Hong Kong, Hong Kong, China. ✉e-mail: magan@ust.hk; Johnny.Chan@cityu.edu.hk

important insight to effectively develop flood management and adaptation strategies.

## Rain rate increase in global landfalling TCs

Based on the TC center locations obtained from the available best-track datasets during the period 1980–2020, we calculate the mean rain rate of all the TCs at each time before they made landfall using the global TC rain rate data from the Multi-Source Weighted-Ensemble Precipitation (MSWEP) product. See Methods on the selection of the landfalling TCs and the extraction of the rain rates. Globally, our observations show that there was a significant increase in the rain rate of the TCs as they approached land (Fig. 1a). During the 60 h before landfall, the mean rain rate of the TCs increases by over 20% from ~1.8 mm h⁻¹ to ~2.2 mm h⁻¹ (Supplementary Table 1). Markedly, the North and South Hemispheres had comparable increased rain rates (Fig. 1b, c). Overall, most of the increase in rain rate originates from the inner core of the TCs (0–200 km radius), while the outer region (200–500 km radius) contributes relatively little (Supplementary Fig. 2a, b). This is because inner-core rainfall is primarily driven by the TC's intrinsic circulation, whereas rainfall in the outer region is more influenced by large-scale environmental conditions[16].

To quantitatively assess the contribution of land effects to enhanced TC precipitation, we further examined changes in the mean intensity of landfalling TCs and their relationship with mean rain rate. Globally, during the 60 h before landfall, TC rain rate and intensity increase at rates of $0.14 \pm 0.01$ mm h⁻¹ per day and $0.29 \pm 0.18$ m s⁻¹ per day, respectively, although the intensity trend is not statistically significant. During the same period, rain rate increases by $0.04 \pm 0.04$ mm h⁻¹ per m s⁻¹ of intensity change. These results indicate that only about 0.012 mm h⁻¹ per day of the observed pre-landfall increase in rain rate can be attributed to TC intensification.

To confirm the results in Fig. 1, we compare the MSWEP rain rates with those from the European Centre for Medium-Range Weather Forecasts (ECMWF) fifth generation atmospheric reanalysis (ERA5) and the Tropical Rainfall Measuring Mission (TRMM) Multi-satellite Precipitation Analysis (TMPA) datasets. Given the relatively short temporal coverage of the TRMM dataset, we only compare TC rain rate records from MSWEP and ERA5 that overlap with the TRMM period. The increasing trend in landfalling TC rain rates is consistently evident across all three datasets (Supplementary Table 1 and Supplementary Fig. 4a). Although satellite- and reanalysis-derived rainfall datasets may exhibit spatial biases in rain rate estimates[17–19], particularly due to differences in observational coverage and algorithm performance between ocean and land regions[20–22], this consistency across independent data sources underscores the robustness and reliability of our finding that TC rain rates increase prior to landfall.

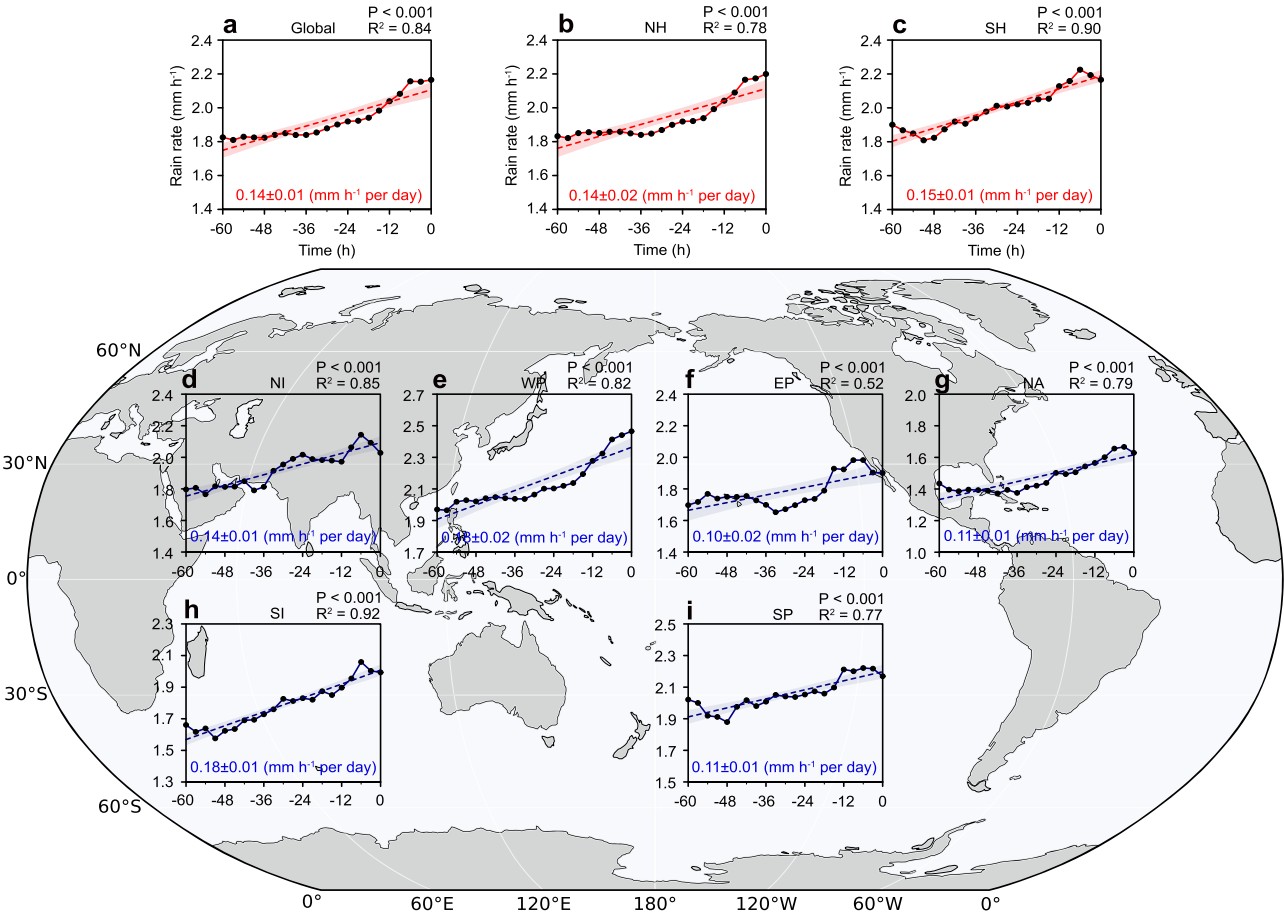

**Fig. 1 | Global changes in rain rate of landfalling tropical cyclones (TCs).** The mean rain rates of all TCs from 60 h before landfall to the time at landfall at 3-hourly intervals (solid line with black dots) and the linear trends (dashed line) from 1980 to 2020. The x-axes time is in hours relative to landfall time (00 h), with negative meaning hours before landfall. The rain rates in different regions are shown by (**a**) Global, (**b**) North Hemisphere (NH), (**c**) South Hemisphere (SH). **d** North Indian (NI), (**e**) Western North Pacific (WNP), (**f**) Eastern North Pacific (ENP), (**g**) North Atlantic (NA), (**h**). South Indian (SI) and (**i**) South Pacific (SP). Statistical significance of the linear trends was assessed using a Student's *t* test. Light shading in all panels represents the two-sided 95% confidence interval. Dashed lines show the linear regression of the mean rain rate with the time before landfall. Regression coefficients (mm h⁻¹ per day) and their standard errors are shown at the bottom center, and the *R*² and *P* values are provided at the top right of each panel.

## Rain rate increases irrespective of TC characteristics

Comparing the rain rates among different oceanic basins shows that the increase in the rain rate of the landfalling TCs exists in all six ocean basins, but the slope of the basin-averaged rain rate varies considerably across the different ocean basins, ranging from 0.10 to 0.18 mm h$^{-1}$ per day. For instance, the southern Indian Ocean has the largest percentage change of ~28%, and the eastern North Pacific has the smallest percentage change with only a ~15% increase. The increase in rain rate of the TCs for the other ocean basins varies from ~15 to ~24 %. We present the summary of the main inter-basin statistics in Fig. 1d–i and Supplementary Table 1.

We further divide the TCs into six latitudinal belts, ranging from 5° to 35° at 5° intervals based on their landfall locations. A significant increasing trend in the rain rate is clearly seen in every belt globally (Supplementary Fig. 4b and Supplementary Table 1). Overall, the increasing trends are larger and more evident near populated regions (10° to 25° global latitude belts), such as the mega-city clusters along the South China coast, where TC landfall effects would be more socially significant. The biggest increasing trend in the global latitudinal belts is in the 15°–20° belt, with a percentage change of 44.52%, whereas the increase in rain rate is less than 10% in the 25°–30° belts. The percentage change of other latitude belts ranges from 10% to 30%.

We also examine the changes in rain rate for each of the TC intensity categories based on the lifetime maximum intensities of the TCs 60 h before landfall at the global scale (see Methods). For all the intensity categories, the landfalling TC rain rates consistently increase, with the regression slope ranging from 0.11 to 0.40 mm h$^{-1}$ per day (Supplementary Table 1 and Supplementary Fig. 4c). The increasing trend is more pronounced for Category 3 TCs (Cat3), with a percentage change of 40.94%, and smallest for Category 2 TCs (Cat2) (12.89%). The increases in the rain rates for the other categories range from ~18% to ~37%. Overall, the increase in rain rate before landfall does not depend on the intensity although the slopes of the regressions vary among the intensity categories.

Overall, our analyses of landfalling TCs at a global scale suggests that rain rates increase significantly as the TCs approach the coast, and all datasets show that this increase continues until landfall, regardless of hemisphere, ocean basin, latitude, or intensity category. It is noteworthy to point out that TCs typically undergo structural and intensity evolution prior to landfall, both of which can influence the rain rate. However, it is difficult to quantify the relative contribution of TC intensity and size to the pre-landfall increase in rain rate, given the limited availability and inherent uncertainty of TC size data, especially near coastlines. Therefore, the relationships between rain rate, intensity, and size, as well as the underlying physical mechanisms driving the short-term variation in TC rain rates, are explored below using numerical experiments.

## Role of land–sea thermal contrasts in the enhancement of TC rain rate

We hypothesize that the increasing trend in the landfalling TC rain rates is associated with enhanced convection, primarily triggered by the contrast between the land and sea. To investigate the hypothesis, we perform a set of idealized simulations of TCs using the Weather Research and Forecasting (WRF4.0) model[23]. In all our experiments, the simulated TCs move from east to west across the domain under a typical 5 m s$^{-1}$ easterly steering wind and a constant sea surface temperature. We configure each experiment with different land surface properties to the west of the initial vortex, with or without frictional differences, and turn on or off the shortwave and longwave radiation schemes. We aim to investigate how the frictional and thermal contrasts between the land and sea affected the TC rain rates. Detailed configurations of the model, the experiments and the physical

parameterization schemes are presented in the Methods and Supplementary Table 2.

Experiment 1 (EXP1, Land-Rad) with land and radiation schemes is the control experiment. The simulated TC rain rate from Land-Rad experiment increases with statistical significance as the TC approaches land (Fig. 2a, b). The slope of the regression is 0.65 ± 0.12 mm h$^{-1}$ per day, which confirms the robustness of our earlier observations. Overall, the simulated TC rain rate increases with both intensity and size during the landfalling phase (Fig. 2c, d). Further analysis of the environmental parameters indicates that as the TC approaches the coast, the low-level relative humidity, convergence, vertical velocity, and instability all increase significantly (Fig. 2e–h; see Methods for details on the extraction of environmental parameters). The land–sea thermal contrast produces distinct diurnal variations in the local land–sea circulations, with stronger daytime warming-induced updrafts and nighttime cooling-induced downdrafts over land compared with those over the sea[24]. This thermal contrast leads to higher low-level humidity over land than over the ocean during the simulation (Fig. 3a–d). As a result of the higher humidity, the water vapor moves from the land to the vortex circulation of TC as it moves closer to land (Fig. 3e). Meanwhile, the difference in the friction between the land and sea generates an asymmetric flow[25], increasing the instability on the offshore side of the TC vortex circulation[26,27] (Fig. 3f). We conclude that the frictional and thermal contrasts between the land and sea lead to increased humidity, strengthened convergent upward motion, and enhanced instability in the TC, all of which favor enhanced asymmetric convection and increased precipitation throughout the affected TC region.

To further investigate the relative importance of land surface roughness and thermal effects on changes in the rain rate of the TC, we conduct two additional experiments. We use the same radiation schemes in Experiment 2 (EXP2, noLand-Rad) as EXP1, but the simulated TC moves westward into a domain without any land surface, thereby removing any differential frictional effects between the land and sea. The simulated TC rain rate in the noLand-Rad experiment increases less than that in the Land-Rad experiment, with a regression slope of 0.34 ± 0.12 mm h$^{-1}$ per day. This smaller increase is accompanied by weaker growth in both TC intensity and size, indicating that land effects not only enhance TC rain rates directly but also indirectly through their influence on TC intensity and size (Supplementary Fig. 5a–d). While the convergent upward motion continues to enhance the TCs development, there is no significant change in the instability, and more importantly, the low-level relative humidity decreases in noLand-Rad experiment (Supplementary Fig. 5e–h). Without the frictional difference between land and sea in noLand-Rad experiment, there is no enhanced asymmetric flow or instability developing on the offshore side of the vortex circulation of the TC. More notably, in the absence of land, the land–sea thermal contrast disappears, producing identical surface properties on both sides of the "imaginary coastline" and preventing the development of local atmospheric circulations. Without these circulations, the transport of near-surface moisture into the mid- to lower troposphere is suppressed, producing a horizontally uniform humidity field (Supplementary Fig. 6). As the TC approaches the "imaginary coastline", the absence of external moisture supply leads to a progressive reduction in mid- to lower-tropospheric humidity in its environment. Even though low-level convergence and upward motion increase, the reduced water vapor content on the offshore side of the vortex still limits the available moisture for convection, being unfavorable for enhanced precipitation in the noLand-Rad experiment.

In Experiment 3 (EXP3, Land-noRad), the model uses the same land surface properties as the Land-Rad experiment, EXP1, but we turn off the radiation schemes. Without radiative effects, the simulated TC rain rate in the Land-noRad experiment shows no increase as the storm

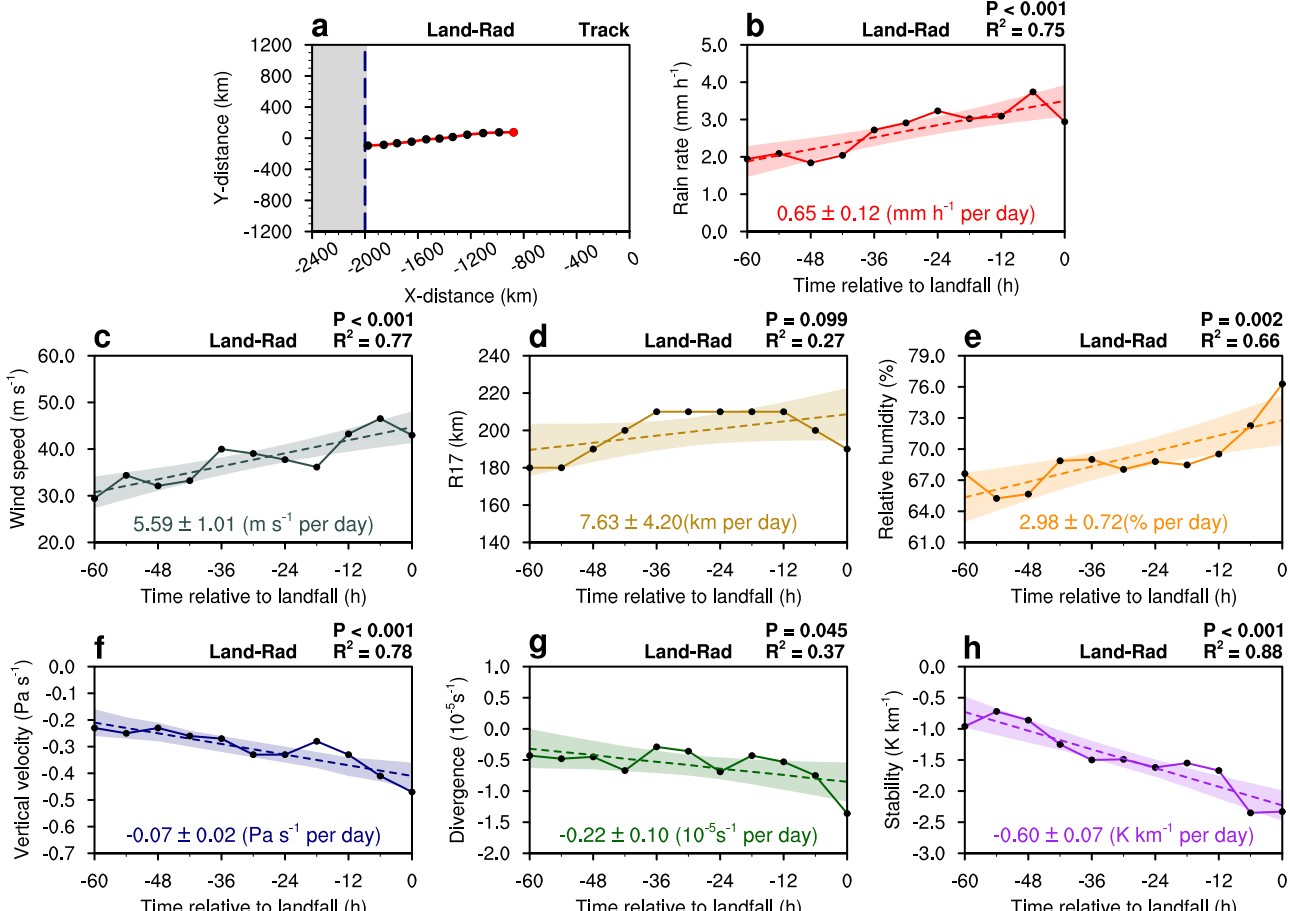

**Fig. 2 | Track and changes in rain rate, intensity, size and environmental parameters of the simulated Tropical cyclone (TC) in the control experiment (EXP1, Land-Rad). a** 60-hr track, (**b**) rain rate (mm h⁻¹), (**c**) maximum wind speed (m s⁻¹), (**d**) radius of 17 m s⁻¹ wind(R17, km), (**e**) relative humidity (%), (**f**) vertical velocity (Pa s⁻¹); (**g**) divergence (10⁻⁵ s⁻¹), and (**h**) stability (K km⁻¹) of the simulated TC at 6-hourly intervals (solid lines with black dots). Different colors of lines and shading represent different environmental parameters across the entire TC (0–500 km radius). In (**a**), the starting point is the TC location 60 h before landfall

(red dot) and the blue dashed line indicates the coastline, with land to the west and the sea to the east. Light shading in (**b–h**) represents the 95% confidence interval of the linear trend. Statistical significance of the linear trends was assessed using a Student's $t$ test. Dashed lines indicate the linear regression of the rain rate, intensity, size and environmental parameters during the time before landfall. Linear regression coefficients with corresponding standard errors are shown at the bottom, and $R^2$ and $p$ values are provided in the top right corner. Time is presented in hours relative to landfall (00 h), with negative values indicating hours before TC landfall.

approaches land, despite the concurrent increases in intensity and size (Supplementary Fig. 7a–d). This behavior is associated with nearly unchanged convergence and vertical velocity, but with noticeable changes in relative humidity and atmospheric stability (Supplementary Fig. 7e–h). In the absence of radiation, no significant humidity contrast exists between land and sea (Supplementary Fig. 8a–d), consistent with the result from the noLand-Rad experiment. Due to the presence of land in Land-noRad experiment, the land-sea frictional differences produce instability similar to that in Land-Rad experiment (Figs. 2h, 3f). However, the Land-noRad experiment has less induced precipitation because of insufficient water vapor and less enhanced convergent upward motion (Supplementary Figs. 8e, 8f).

To summarize, we demonstrate that land-sea differences in surface friction and thermal effects lead to increases in low-level relative humidity, convergence, vertical velocity, and instability within the TC environment, all of which contribute to the enhanced asymmetric convection and increased rain rate of landfalling TCs. Figure 4 summarizes the various physical processes with a schematic diagram. Key controlling factors are the frictional and thermal contrasts between land and sea that influence the increasing trend in rain rate and the enhanced asymmetric convection, with the thermal contrasts being the most significant. In addition to the influence of land–sea thermal contrast on TC rainfall, land–sea frictional heterogeneity can induce

asymmetric circulation and convection in landfalling TCs. Such asymmetries may generate localized potential vorticity anomalies, which can accelerate storm translation. The change in TC translation speed may further modulate regional rain rates.

## Discussion

This study highlights the short-term variation in the precipitation of landfalling TCs and the underlying physical processes; a result which has global implications for hazard prevention. The importance of understanding the change in TC precipitation as the storm approaches a coastline on a global scale lies in the significant increase in the rain rate which occurs because of the physical differences between the land and sea regardless of the TC's location in the world. We find short-term increases in rain rate associated with increased relative humidity and enhanced convection in the vortex circulation of the TC, which are triggered by the land-sea contrasts under a constant sea surface temperature condition. Unlike many earlier studies that suggest the *long-term* increase in rain rates is associated with rising sea surface temperatures due to global warming[8,28], we emphasize that the short-term increase in rain rate due to land-sea contrasts should also be taken into consideration under a warming climate[8]. The short-term effect on the rain rate has the combined potential to cause severe flooding and more intense storm surges in coastal regions[29] because of the overall

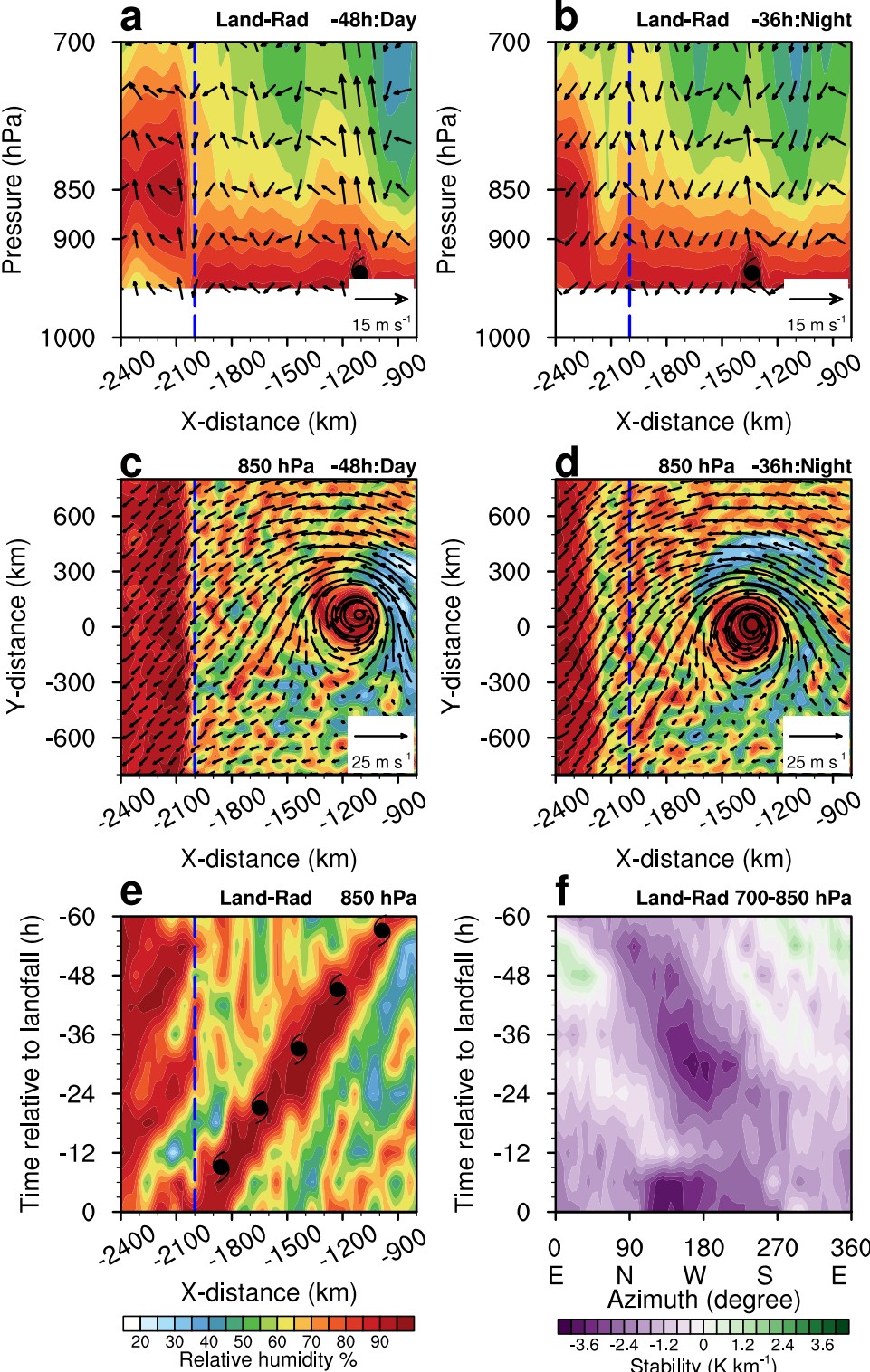

**Fig. 3 | Spatial and temporal variations of wind vectors (m s⁻¹), relative humidity (shading, %), and stability (K km⁻¹) in experiment 1 (EXP1, Land-Rad).** **a**, **b** Radius–pressure Hovmöller diagrams of wind and relative humidity at $y = 0$ km for the TC during the 48 h of daytime and 36 h of nighttime before landfall. **c**, **d** horizontal distributions of wind and relative humidity fields at 850 hPa during daytime and nighttime. **e**, **f** azimuthal–time Hovmöller diagrams of respective relative humidity at 850 hPa and stability (the difference in equivalent potential temperature between 850 hPa and 700 hPa). In (**a**–**d**), the time in the top left corner is in hours relative to landfall (00 h), with negative values indicating hours before landfall. Day and night refer to horizontal and vertical distributions of wind vectors and relative humidity (shading) during the day and night, respectively. In (**e**, **f**), we averaged the azimuthal–time Hovmöller diagram over the tropical cyclone radius from 0 to 500 km. The blue dashed line indicates the coastline, with land to the west and the sea to the east. In (**a**–**e**), the radius values on the x- and y-axes are presented relative to the model domain center, with negative values indicating positions west (south) of the domain center and positive values indicating positions east (north) of the domain center. In (**a**, **b**, **e**), the typhoon symbol corresponds to the x-axis values of the TC center location. In (**f**), the letters E, N, W, and S represent East, North, West, and South in Earth's coordinate system.

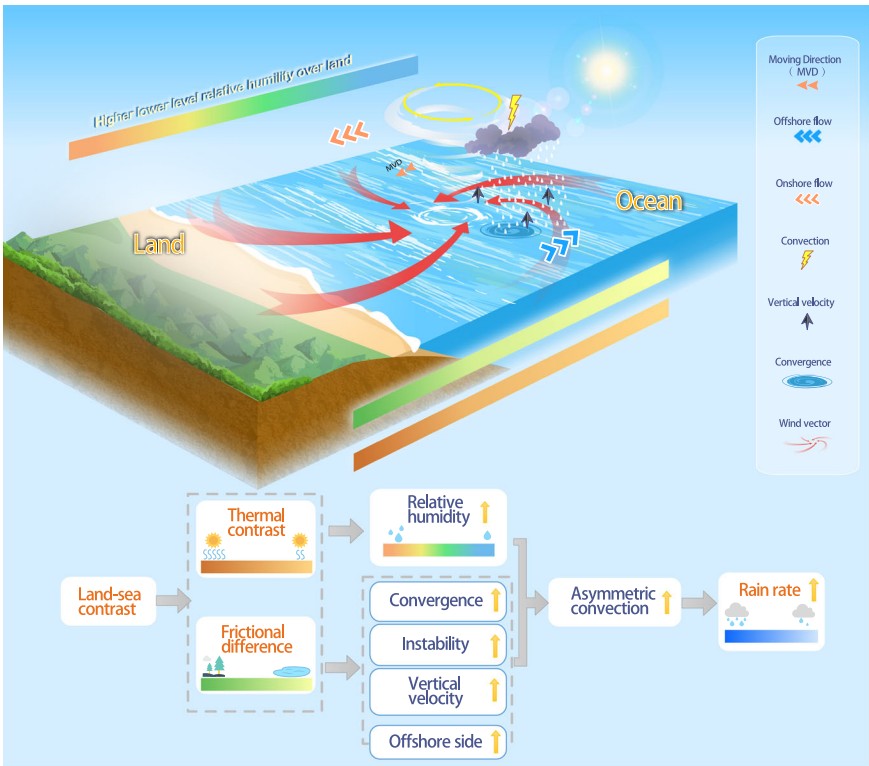

**Fig. 4 | Schematic diagram of the physical mechanisms for the increase in rain rate of landfalling Tropical cyclone (TC).** The white spiral and yellow circle represent the simplified structure of the landfalling TC in North Hemisphere and TC's vortex circulation, respectively. The orange vector (solid arrow) in front of the TC center indicates the moving direction (MVD) of the TC. Red curved arrows represent the vortex circulation of the TC (wind vector). Black upward arrows represent the vertical velocity of the TC. The cloud represents the asymmetric convection on the offshore side of the vortex circulation that contributes to the precipitation (white dashed lines). Blue and orange vectors represent the offshore (from land towards ocean) and onshore (from ocean towards land) flow respectively. The rainbow-colored bar, green bar, and yellow bar represent the differences in relative humidity, thermal differences, and frictional differences between the ocean and land, respectively.

greater accumulated precipitation associated with the landfalling TCs at any time scale.

Over the past 50 years, TCs have claimed close to 800,000 lives globally and resulted in a total direct economic loss of US $1.4 trillion[30]. The detrimental effects of TCs on humans depends on the frequency, intensity, and precipitation of the storms and, especially, on the number of people exposed to the storms and the human vulnerability[31]. As urban centers grow and coastal populations rise, the increased rain rates of landfalling TCs will lead to even greater economic losses and casualties in coastal regions worldwide, where approximately 560 million people are exposed to these storms[32]. Our findings significantly enhance the current understanding of changes in the rain rate of global TCs before they reach land and have critical implications for flood management and adaptation strategies, but there is potential for furthering the study of TC rain rates.

Globally, urbanization alters land surface properties significantly. For example, there are many highly urbanized coastlines in parts of east Asia with surface properties very different from the more rural coastal areas. The orientation of the coastlines also varies for different parts of the world. These factors are key elements of the land-sea thermal contrast associated with TC rain rates. We therefore need to further analyze observations and conduct additional experiments using diverse coastline orientations and different types of land use. Accordingly, the numerical weather prediction models for the prediction of rainfall distributions associated with TC landfall must represent land surface properties accurately so that the proper physical processes can be predicted.

## Methods
### Data
We use several types of datasets in this study: TC best-track data, global precipitation products, and atmospheric reanalysis. We use the TC best-track information provided by the International Best Track Archive for Climate Stewardship (IBTrACS, v0400)[33] from 1980 to 2020, including global TC records from various agencies. We use records from the World Meteorological Organization (WMO) that contain records on the latitude and longitude of TC locations, maximum wind speed (MWS), landfall and dist2land (distance to the nearest land) at 3-hourly intervals.

We extract the global TCs precipitation data from the most recent available Multi-Source Weighted-Ensemble Precipitation (MSWEP) dataset from 1980 to 2020. MSWEP combines rain gauge, satellite, and reanalysis data to produce high-quality rainfall estimates at 3-hourly intervals with 0.1° latitude and longitude resolutions[34]. Precipitation of MSWEP dataset has been widely used because it is a long-term, continuous record with higher quality and high resolution in densely gauged and ungauged regions[35]. We also use the rain rate from the Tropical Rainfall Measuring Mission (TRMM) TMPA 3B42 product[22], and the total precipitation from the fifth-generation European Centre for Medium-Range Weather Forecasts (ECMWF) atmospheric reanalysis (ERA5) dataset to verify our results[36]. The TRMM and ECMWF data provide 3-hourly intervals and a 0.25° spatial resolution for their precipitation estimates. For consistency, we convert the total precipitation from the ERA5 dataset to rain rate at 3-hourly intervals.

In this study, we primarily use the MSWEP dataset from 1980 to 2020, rather than the TRMM dataset from 1999 to 2019 because the

MSWEP dataset aligns with the TC best-track records from the post-satellite era, which are generally more homogeneous and reliable, and the MSWEP data provide us with enough samples to study. Once more, for consistency, we only use records of TC rain rates from the MSWEP dataset from 1999 to 2019 that overlap with the same period as the TRMM dataset when we validate the robustness of our observed results.

### Selecting the landfalling TCs

We use the records of latitude, longitude, wind speed, landfall, and dist2land from the IBTrACS data to identify the landfalling TCs. The dist2land provides the distance to the nearest landfall point. A value of zero means the TC was over land, and the landfall time with a value of zero means that the TC crossed a coastline before the next record. We define landfall = 0 and dist2land = 0 as TC landfall. We define two selection criteria for landfalling TCs. Firstly, we only consider landfalling TCs with a lifetime maximum intensity of at least 35 knots and a lifespan of more than 60 h before landfall. Our focus was on the stage within the 60 h before landfall, because this period is the most crucial for disaster mitigation, prevention, and preparedness. During this 60-h stage, land surface properties gradually influence TCs, making those 60 h essential for accurate forecasting and mitigation efforts. Secondly, we filter out the TCs that underwent an extra-tropical transition.

With these two criteria, we select a total of 1,468 landfalling TCs during the 41-year period from 1980 to 2020. Based on the latitude and longitude of the landfall locations, we classify all the TCs into different geographic categories as being global or in different hemispheres (southern and northern hemispheres), basins (WNP: western North Pacific, ENP: eastern North Pacific, NA: North Atlantic, SI: South Indian Ocean, SP: South Pacific, NI: North Indian Ocean), latitudinal belts (≤10°, 10°–15°, 15°–20°, 20°–25°, 25°–30°, 30°–35°), and the Saffir-Simpson Hurricane Wind Scale (TS, Cat 1, Cat 2, Cat 3, Cat 4, Cat 5).

### Identifying the precipitation related to landfalling TCs

For each landfalling TC, we use the latitude and longitude of the center location at 3-hourly intervals to identify the TC rainfall from various precipitation products. Following the most common method[8,9], we define the TC rainfall as the rainfall within a fixed radius of 500 km from the storm center, corresponding to the estimated radius where the mean rain rate exceeds $0.5 \, mm \, h^{-1}$ for global TCs[37]. Similarly, we define the TC inner core as the region within a radius of 0 to 200 km and the outer region as the region between 200 and 500 km. After extracting the TC rainfall, we project the grid precipitation as a function of the radial distance in 10-km intervals and the azimuth in 10° intervals relative to the storm center. Correspondingly, we construct the radius–time Hovmöller plots of the all-pixels average rain rate (with non-rainy pixels counted as zero) and the conditional rain rate (counting only rainy pixels with a value exceeding $0.1 \, mm \, h^{-1}$) (Supplementary Fig. 1). The distributions of conditional and average rain rates are well-aligned. Because the conditional rain rate captures the realistic precipitation distribution better[38], we use the conditional rain rate to construct the time series of the average rain rate at 3-hourly intervals from the 60 h before landfall until the time of landfall. If not otherwise specified, the 'average rain rate' for all global TCs, for the two hemispheres, and for the other groupings, refers to the averaged conditional rain rate of the entire TC. Moreover, we also use the method described above to identify various variables of the TCs from the ERA5 dataset and to calculate the time series of all-pixels averaged values of each variable for the entire TC.

### Statistics

We examine the trends in the mean rain rate of all the TCs or each group of TCs using linear regression with the full degrees of freedom in the time series determining the $p$ value for the regression coefficients and correlation coefficient. To assess the robustness of the regression, we used the two-tailed Student's $t$ test. Dividing the difference between the final and initial points of the regression line by the value of the initial point determines the percentage change of the rain rate.

### Configuring the model and designing the experiments

We use the Weather Research and Forecasting (WRF) version 4.0[23] to simulate an ideal TC on the double periodic $f$ plane. We set the constant Coriolis parameter, $f$, to $5.0 \times 10^{-5} \, s^{-1}$, centered at 20°N, and a constant sea surface temperature (SST) of 28.5 °C throughout the simulation. The model adopts a double-nested configuration with horizontal grid spacing of 20 km (outermost domain size: $12000 \times 12000$ km) and 10 km (innermost domain size: $6000 \times 6000$ km). In addition, the model has 26 vertical levels, and the model top is set at 25 km. Our integration time step is 40 seconds. Physical parameterizations include the Yonsei University boundary layer scheme, the surface layer scheme by Dudhia et al [39], the WRF single-moment six-class microphysics[40], and the Kain-Fritsch (new Eta) cumulus scheme[41]. Specifically, our choices of using the RRTMG shortwave[42] and longwave radiation schemes[43] depend on our experiment designs (Supplementary Table 2).

Similar to Tu et al.[37], we initialize the model at the domain center with an axisymmetric vortex in hydrostatic and gradient wind balance, with a maximum tangential wind speed of $15 \, m \, s^{-1}$ at a radius of 82.5 km and an outer radius of 412.5 km where the tangential wind speed drop to zero. We derive the basic-state potential temperature and humidity used for initializing the vortex from the mean hurricane season sounding of Jordan[44]. This has been the most common model configuration used for ideal simulations in previous studies[12,45].

To investigate the physical processes influencing precipitation changes in landfalling TCs, we conduct three numerical experiments. In all the simulations, we subject the initial axisymmetric vortex to a $5 \, m \, s^{-1}$ easterly steering flow ($u = 5 \, m \, s^{-1}$, $v = 0 \, m \, s^{-1}$), directing it towards the western side of the domain. The magnitude of the steering flow is comparable to the usual translation speed of landfalling TCs globally[5]. To isolate the respective roles of the land surface and radiation, three experiments are named Land-Rad, noLand-Rad, and Land-noRad. In Experiment 1 (EXP1, Land-Rad), referred to as the control experiment, we use the radiation schemes, and the domain configuration includes a combination of rectangular sea and land surfaces. Specifically, the model coastline is configured as a north-south oriented boundary with dryland, cropland, and pasture surface categories, positioned 1000 km east of the western boundary of the innermost domain. Consequently, even at the time of landfall, the TC center remains over 1000 km from the lateral boundary, a distance that well exceeds the 500 km radius used for rain-rate averaging. Thus, the lateral boundary is considered unlikely to introduce meaningful impacts on the simulated TC structure or the calculated mean rain rate. In Experiment 2 (EXP2, noLand-Rad), to investigate the role of land surface friction in the changing landfalling TC precipitation, we only use the radiation schemes, and the domain configuration in this experiment does not include land. Experiment 3 (EXP3, Land-noRad) has the same domain configuration as the control experiment but without any radiation schemes so that we can assess the effect of the land-sea thermal contrast on landfalling TC precipitation. We integrate all the simulations for 240 h, with variables output at 6-hourly intervals on the model's eta levels and then interpolate to pressure levels. Because our main interest is to determine the role of land-sea contrast in changing landfalling TC precipitation, we extract model outputs from 60 h before landfall to the time of landfall. Landfall time is defined as the moment the TC crosses a north–south oriented coastline in the Land-Rad and Land-noRad experiments. In the noLand-Rad experiment, where no actual coastline exists, landfall time is set to match that of the Land-Rad experiment. We determine the TC track based on the minimum center pressure at the surface, and we use the difference in accumulated total precipitation between the current and next time steps to calculate the rain rate.

## Atmospheric stability

Atmospheric stability that accounts for both temperature and moisture is quantified by moist static stability (MSS). It is defined as the difference in equivalent potential temperature $\theta_e$ between 700 and 850 hPa[26], which is given by

$$\text{MSS} = \theta_{e\,700} - \theta_{e\,850} \qquad (1)$$

The equivalent potential temperature[46] $\theta_{se}$ is defined as:

$$\theta_e = T \left( \frac{P_0}{P} \right)^{\frac{R_d}{c_p}} \exp \left( \frac{L_v q}{R_d T} \right) \qquad (2)$$

where $T$, $q$ and $P$, are the absolute temperature ($K$), the specific humidity (kg/kg), and the atmospheric pressure (hPa), respectively. $R_d$ is the gas constant for dry air, and $c_p$ is the specific heat capacity at a constant pressure. Typically, $\frac{R_d}{c_p} = 0.286$ for air. $P_0$ is the reference pressure, usually taken as 1000 hPa, and $L_v$ is latent heat of vaporization, with a constant value of $2.5 \times 10^6$ J/kg.

## Data availability

The International Best Track Archive for Climate Stewardship (IBTrACS, v0400) dataset can be obtained from the National Centers for Environmental Information (available online at https://www.ncei.noaa.gov/products/international-best-track-archive). TC-related precipitation data are obtained from the Multi-Source Weighted-Ensemble Precipitation (MSWEP) dataset (https://www.gloh2o.org/mswep/), the Tropical Rainfall Measuring Mission (TRMM) TMPA 3B42 product (https://climatedataguide.ucar.edu/climate-data/trmm-tropical-rainfall-measuring-mission), the fifth-generation European Centre for Medium-Range Weather Forecasts (ECMWF) atmospheric reanalysis (ERA5) dataset (https://cds.climate.copernicus.eu/datasets?q=era5). The processed source data used to perform the analyses and generate all figures have been archived at Zenodo[47] (https://doi.org/10.5281/zenodo.17658283).The model output data is too large to be deposited in a public repository, but they are available from the first author (Q.Z., email: zqj@ust.hk) upon request. Source data are provided with this paper.

## Code availability

The data analysis and visualization were performed using the NCAR Command Language (NCL). In particular, the diagnostic analysis follows the procedures described in the Methods section and relies on customized scripts that involve multiple function libraries and complex workflows. Because these codes cannot be packaged into a stable and publicly shareable form, they are available from the first author (Q.Z., email: zqj@ust.hk) upon request.

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

## Acknowledgements

This research was supported by the Area of Excellence Scheme, Hong Kong Research Grants Council (Grant/Award Number: AoE/P-601/23-N) and by the Center for Ocean Research in Hong Kong and Macau (CORE). CORE is a joint ocean research center between the Laoshan National Laboratory for Marine Science and Technology and the Hong Kong University of Science and Technology. S.T. was supported by the Guangdong Basic and Applied Basic Science Research Foundation (2024A1515010714). R.T. was supported by the Natural Environment Research Council (NE/W009587/1) and the Singapore Green Finance Centre. We thank National Supercomputing Center in Guangzhou Nansha Sub-center for computational resources.

## Author contributions

Q.Z., J.G., and J.C.L.C. conceived the idea and designed the investigation. Q.Z. conducted the data analyses, designed the numerical experiment, and wrote the initial manuscript. J.G and J.C.L.C. co-wrote the manuscript. S.T. and R.F. contributed to the discussion and interpretation of the results. All authors contributed to the final manuscript.

## Competing interests

The authors declare no competing interests.
