## [Transparent Peer Review file · Nature Communications]

Global Increase in Rain Rate of Tropical Cyclones Prior to Landfall

Corresponding Author: Professor Jianping Gan

Version 0:

Reviewer comments:

Reviewer #1

(Remarks to the Author)

See attached

Reviewer #2

(Remarks to the Author)

Review of "Global Increase in Rain Rate of Tropical Cyclones Prior to Landfall" by Zhong et al. for Nature Communications (#NCOMMS-25-51671)

Recommendation: Minor Revision

General comments:

Using 40-yr global rainfall data, this study found that rain rates of tropical cyclones (TCs) increase significantly as the TCs approach the coast from 60 h prior to landfall to the landfall time, and all datasets show the increase continues until landfall, regardless of hemisphere, ocean basin, latitude, or TC intensity category. It was hypothesized that the increasing trend in the landfalling TC rain rates is associated with enhanced convection, primarily triggered by the contrast between the land and sea. To investigate the hypothesis, a set of idealized simulations of TCs using the Weather Research and Forecasting (WRF4.0) model was performed. It was found that land-sea differences in surface friction and thermal effects lead to increases in low-level relative humidity, convergence, vertical velocity, and instability within the TC environment, all of which contribute to the enhanced asymmetric convection and increased rain rate of landfalling TCs.

The paper is well-written and easy to follow. The results are interesting. Although the general phenomenon of rain rate increasing as TCs approach to coast/land from ocean has been shown by previous studies (Jiang et al. 2008 and Guzman and Jiang 2023, see my minor comment below), the authors did a great job in presenting this increase as a function of time from 60 h prior to landfall to the landfall time. I also applaud the authors' effort of testing the hypothesis using idealized simulations. The physical explanation of the short-term increasing of TC rainfall prior to landfall is scientifically sound and important for advancing our knowledge in the field. I only have one minor comment below. I believe that the paper is suitable for publishing in Nature Communications after addressing the comment.

Minor Comment:

Line 68-69: Please add the following two references here. An early study by Jiang et al. (2008) using TRMM TMPA data showed that TC rain rates over land and coast are larger than those over ocean (their Fig. 1). More clearly, using 20 years of IMERG satellite rainfall data for global TCs, Guzman and Jiang (2023)'s Fig. 2 showed an increase of TC rain rates from ocean to off-shore then to near-land stages, very similar to what was found in this study.

Jiang, H., J. B. Halverson, J. Simpson, and E. J. Zipser 2008: Hurricane "Rainfall Potential" Derived from Satellite Observations Aids Overland Rainfall Prediction. *J. Appl. Meteor. Climatol.*, 47, 944-959.

Version 1:

Reviewer comments:

Reviewer #1

(Remarks to the Author)

Thank you for the authors' detailed responses. However, despite their clarifications, several key concerns remain unresolved and, in some respects, have become more significant.

My main point in the previous comments was based on the well-established relationship that TC rain rate increases with TC intensity. Therefore, the observed increase in rain rate as TCs approach land could partly result from the concurrent intensification of TCs rather than land effects alone. In the idealized experiments, the authors clearly demonstrated that even without land, rain rate increases as TC intensity increases. While I agree that land interaction can further enhance rainfall, it is essential to discuss how much of the rain rate increase shown in Figure 1 is attributable to intensity changes. To evaluate this, I suggest plotting the same sample data from Figure 1 using TC intensity instead of rain rate. If intensity increases along the same timeline, it would indicate that part of the rain rate increase is indeed due to TC strengthening.

Additionally, when TCs move poleward from lower to higher latitudes, they encounter cooler and drier air masses, which can often enhance precipitation due to increased baroclinicity. Thus, rain rate may also increase with latitude. It would therefore be informative to calculate the mean latitude of the TCs in Figure 1. If latitude increases alongside rain rate, this would suggest that TC position (latitude change) also contributes to the rain rate increase.

I also believe that non-landfalling TCs can show rain rate increases associated with changes in intensity and latitude. The authors should carefully analyze how these factors affect rain rate for non-landfalling cases. Without such analysis, it is difficult to attribute the observed rain rate increase in landfalling TCs solely to land effects.

The additional Figure A1 presented by the authors does not adequately address this concern, as it does not include information on TC intensity or latitude. I strongly encourage the authors to provide an additional analysis showing how TC intensity and latitude variations influence rain rate in non-landfalling TCs.

Regarding the comparison between EXP1 and EXP2, I also have several questions:

1) It is generally understood that as a TC intensifies, mid- to lower-tropospheric relative humidity increases due to enhanced convection, condensation, and latent heat release. However, in EXP2, TC intensity increases while relative humidity does not. Could the authors explain this discrepancy?

2) The TC track comparison between EXP1 and EXP2 shows that the TC in EXP2 moves considerably more slowly, despite both experiments using an easterly steering flow of 5 m s^{-1} . What causes such a large difference in translation speed? This discrepancy could significantly influence TC structure and rain rate. Could the experimental design be modified to minimize the effects of such translation speed differences?

Finally, I am particularly concerned about the limited model domain size. In EXP1, when the TC reaches land, its center is only about 400 km from the domain boundary. Given that the rain rate is averaged within a 500 km radius, the outer 100 km portion would be excluded, potentially artificially inflating the mean rain rate near landfall. The authors should carefully assess and discuss how this boundary limitation might affect the quantitative results and conclusions.

Version 2:

Reviewer comments:

Reviewer #1

(Remarks to the Author)

The authors were successfully addressed all major comments raised by me and therefore I accept this work to be published in *Nature Communications* in the present form.

Responses to Referee

Manuscript number: NCOMMS-25-51671

Title: Global Increase in Rain Rate of Tropical Cyclones Prior to Landfall

Author(s): Quanjia Zhong et al.

Revision on October 30, 2025

Responses to Reviewer #1

Overall summary of the main contribution of the paper

This study investigates short-term variations in tropical cyclone (TC) rainfall, particularly around landfall. Using global satellite data, the authors find that TC mean rain rates increase by more than 20% from 60 hours before landfall to the time of landfall, regardless of basin, hemisphere, or intensity. The enhancement is linked to land–sea thermal contrasts, increased low-level humidity over land, and frictional convergence offshore, which together intensify convection and precipitation.

This manuscript is engaging, clearly written, and makes an excellent contribution in terms of scientific importance and societal relevance. In particular, understanding rainfall processes near TC landfall is highly valuable for disaster risk reduction. However, before publication, additional analysis and discussion regarding other factors influencing the increase in rain rate are necessary.

Response: Thanks to the reviewer's time and valuable suggestions for the improvement of the manuscript. We revised the manuscript carefully to the reviewer's comments. More details and point-to-point responses to the reviewer's comments are listed as follows (reviewers' comments in **black font**, our replies in **blue font**). Particularly, the **red font** in the manuscript indicates some important changes in comparison with the previous version.

Main Questions/Comments

My detailed comments are as follows:

- 1. Many TCs typically undergo structural development prior to landfall. During this period, intensity and size (e.g., R17, radius of 17 m s⁻¹ wind) often increase, which can directly influence rain rate. Consequently, part of the observed 60-hour increase in rain rate may be attributable to changes in intensity and size, independent of landfall itself. Although quantifying this contribution may be challenging, the authors should **at least discuss how such changes occur within the 60-hour window** and evaluate their potential impact on the observed rain rate increase. **It would also be valuable to examine whether non-landfalling TCs exhibit similar rain rate increases, since intensification and size growth alone could enhance precipitation without land interaction.** For non-landfalling TCs, one could determine the time*

interval (TI) between the lifetime maximum intensity (LMI_time) and landfall for landfalling TCs, and then apply this mean TI to the non-landfalling cases. By adding this mean TI to each non-landfalling TC's LMI_time, an equivalent reference time can be established, allowing the rain rate during the preceding 60 hours to be calculated. This would enable a fairer comparison between landfalling and non-landfalling TCs.

Response:

We thank the reviewer for this insightful comment. We agree that TCs typically undergo structural and intensity evolution prior to landfall, both of which can influence the rain rate. As the reviewer correctly noted, it is difficult to quantify the relative contribution of TC intensity and size to the pre-landfall increase in rain rate, given the limited availability and inherent uncertainty of TC size data. This is especially the case for TCs near coastlines because observational data for the part of a TC over land are likely either not available or very uncertain. To address this issue, we attempt to evaluate the potential impacts of TC intensity and size on rain rate using sensitivity numerical experiments, as described in our response to Major Comment 2. Meanwhile, we have also added a brief discussion in the subsection titled “Rain rate increases irrespective of TC characteristics” in the revised manuscript to clarify this point further. The relevant details are as follows:

“It is noteworthy to point out that TCs typically undergo structural and intensity evolution prior to landfall, both of which can influence the rain rate. However, it is difficult to quantify the relative contribution of TC intensity and size to the pre-landfall increase in rain rate, given the limited availability and inherent uncertainty of TC size data, especially near coastlines. Therefore, the relationships between rain rate, intensity, and size, as well as the underlying physical mechanisms driving the short-term variation in TC rain rates, are explored below using numerical experiments.”

Regarding the suggestion to compare landfalling and non-landfalling TCs, we believe that such a comparison could be affected by sampling differences between the two TC groups, rather than land interaction itself. Since our primary objective is to investigate landfalling TCs, our dataset naturally focuses on those that actually make landfall.

In line with the reviewer's suggestion, however, we further examined the rain-rate evolution of landfalling TCs as a function of their distance to land. This approach avoids the sampling bias between different TC populations and allows the direct identification of land effects. As shown in Figure A1 (Extended Data Fig. 3 in revised manuscript), when a TC is more than 800 km away from land, it is mainly over the open ocean without any influence from land, and its rain rate does not change significantly. In contrast, within 800 km of land, the rain rate increases markedly as the TC approaches the coast. This result is consistent with the result shown in Figure 1 of the main text. From 60 hours before landfall to the landfall time (0 hours), the rain rate exhibits a continuous and significant rise, likely influenced by land-induced effects. These results reinforce the robustness of our main finding that the TC rain rate tends to increase as TCs approach land.

We appreciate the reviewer's constructive suggestions. The new Extended Data Fig. 3 and corresponding text describing the two-phase comparison have been added to the revised manuscript. The relevant details are as follows:

“To further assess the effect of land, we examined the changes in rain rate of landfalling TCs as a function of their distance to land (Extended Data Fig. 3). When a TC is more than 800 km away from land, it is over the open ocean without any land influence, and its rain rate shows no significant change. However, within 800 km of land, the rain rate increases markedly as the TC approaches the coast, likely due to land effects. This pattern, consistent with the time-based analysis shown in Fig. 1, clearly indicates a negligible trend in the former phase but a significant increase in the latter. These results reinforce the robustness of the observed pre-landfall increase in TC rain rate and highlight the possible contribution of land-induced processes.”

Figure A1 Rain rate of tropical cyclones (TCs) as a function of distance to land.

The mean rain rate of all TCs is shown at 25 km intervals from 1600 km to 0 km from land (solid line with black dots), along with their linear trends (dashed lines) for the two phases: TCs more than 800 km away from land and those within 800 km of land. Light shading represents the two-sided 95% confidence range of the linear trend. The linear regression coefficients with uncertainty estimates (unit: mm h⁻¹ per 800 km) are indicated at the bottom, and the corresponding R² and p values are shown at the top. Distance is expressed in kilometers relative to land (0 km), with positive values indicating positions prior to landfall. The vertical black dashed line marks the distance of 800 km from landfall.

2. *The same concern applies to the numerical experiments. As TCs propagate, their intensity and size typically increase, which could contribute to enhanced rain rates. This may partly explain the rain rate increase observed even in EXP2 (without land included). The fact that EXP1 (with land) shows a greater increase than EXP2 suggests that this excess can be attributed to land effects. However, to interpret this robustly, the manuscript **should present how TC characteristics (intensity and size) evolve in both experiments**. For instance, if land interaction alters intensity or size, this could also contribute to rain rate changes, and such effects should be disentangled from the pure land–sea contrast mechanism.*

Response:

We appreciate this valuable comment. Following the reviewer’s suggestion, we have examined the evolution of TC intensity and size (radius of 17 m s⁻¹ wind) in both numerical experiments (EXP1: Land-Rad and EXP2: noLand-Rad). The results

show that both intensity and size generally increase during the 60-hour period, and the rates of increase are greater in EXP1 than in EXP2 (see Figures A1 and A2). This indicates that land–sea contrasts may **not only directly enhance TC rain rates** but **may also indirectly contribute through modifications of TC intensity and size.**

We have added these analyses and corresponding discussions in the Results section of the revised manuscript to clarify this mechanism. Figure 2 and Extended Data Figures. 5 and 7 have been replotted with additional analyses of TC intensity and size. The relevant details are as follows:

“The simulated TC rain rate in the noLand-Rad experiment increases less than that in the Land-Rad experiment, with a regression slope of $0.34 \pm 0.12 \text{ mm h}^{-1}$ per day. This smaller increase is accompanied by weaker growth in both TC intensity and size, indicating that land effects not only enhance TC rain rates directly but also indirectly through their influence on TC intensity and size (Extended Data Figs. 5a–d).”

Figure A2 Track and variations in rain rate, intensity, and size of the simulated TC in the control experiment (EXP1, Land-Rad). (a) 60-hour track, (b) rain rate (mm h⁻¹), (c) maximum wind speed (m s⁻¹), and (d) radius of 17 m s⁻¹ wind (R17, km) of the simulated TC at 6-hour intervals (solid lines with black dots). Different line colors and shaded areas represent different variables. In panel (a), the starting point

denotes the TC location 60 hours before landfall (red dot). Light shading in panels (b)–(d) indicates the 95% confidence interval of the linear trend. Dashed lines denote the linear regression of rain rate and environmental parameters before landfall. Linear regression coefficients with error estimates are shown at the bottom, and R^2 and p values are provided in the top-right corner. Time is given in hours relative to landfall (0 h), with negative values indicating hours before TC landfall.

Figure A3 As in Fig. A2, except for Experiment 2 (EXP2, noLand-Rad).

Responses to Reviewer #2

General comments:

Using 40-yr global rainfall data, this study found that rain rates of tropical cyclones (TCs) increase significantly as the TCs approach the coast from 60 h prior to landfall to the landfall time, and all datasets show the increase continues until landfall, regardless of hemisphere, ocean basin, latitude, or TC intensity category. It was hypothesized that the increasing trend in the landfalling TC rain rates is associated with enhanced convection, primarily triggered by the contrast between the land and sea. To investigate the hypothesis, a set of idealized simulations of TCs using the Weather Research and Forecasting (WRF4.0) model was performed. It was found that land-sea differences in surface friction and thermal effects lead to increases in low-level relative humidity, convergence, vertical velocity, and instability within the TC environment, all of which contribute to the enhanced asymmetric convection and increased rain rate of landfalling TCs.

The paper is well-written and easy to follow. The results are interesting. Although the general phenomenon of rain rate increasing as TCs approach to coast/land from ocean has been shown by previous studies (Jiang et al. 2008 and Guzman and Jiang 2023, see my minor comment below), the authors did a great job in presenting this increase as a function of time from 60 h prior to landfall to the landfall time. I also applaud the authors' effort of testing the hypothesis using idealized simulations. The physical explanation of the short-term increasing of TC rainfall prior to landfall is scientifically sound and important for advancing our knowledge in the field. I only have one minor comment below. I believe that the paper is suitable for publishing in Nature Communications after addressing the comment.

Response:

We thank the reviewer for his/her helpful suggestions, which helps improving this manuscript. In revised paper, we have addressed the major concern about the prior work and carefully revised the manuscript according to the comments and suggestions raised by the reviewer. Please find below a detailed point-by-point response to all comments (reviewers' comments in **black font**, our replies in **blue font**). In addition, the **red font** in the revised manuscript indicates some changes in comparison with the previous version.

Minor Comment:

1. *Line 68-69: Please add the following two references here. An early study by Jiang et al. (2008) using TRMM TMPA data showed that TC rain rates over land and coast are larger than those over ocean (their Fig. 1). More clearly, using 20 years of IMERG satellite rainfall data for global TCs, Guzman and Jiang (2023)'s Fig. 2 showed an increase of TC rain rates from ocean to off-shore then to near-land stages, very similar to what was found in this study.*

Jiang, H., J. B. Halverson, J. Simpson, and E. J. Zipser 2008: Hurricane "Rainfall Potential" Derived from Satellite Observations Aids Overland Rainfall Prediction. *J. Appl. Meteor. Climatol.*, 47, 944-959.

Guzman, O. and H. Jiang, 2023: Climatology of Tropical Cyclone Rainfall Magnitude at Different Landfalling Stages: An Emphasis on After Landfall Rain. *J. Appl. Meteor. Climatol.*, 62, 801–815. <https://doi.org/10.1175/JAMC-D-22-0055.1>.

Response:

We thank the reviewer for the helpful comment. Relevant references have been added to the Introduction section and the reference list in the revised manuscript.

Responses to Referee

Manuscript number: NCOMMS-25-51671A

Title: Global Increase in Rain Rate of Tropical Cyclones Prior to Landfall

Author(s): Quanjia Zhong et al.

Revision on November 16, 2025

Responses to Reviewer #1

Overall summary of the main contribution of the paper

Thank you for the authors' detailed responses. However, despite their clarifications, several key concerns remain unresolved and, in some respects, have become more significant.

Response: We sincerely thank the reviewer for the time and thoughtful comments provided. We have carefully revised the manuscript in response to all concerns. More details and point-to-point responses to the reviewer's comments are listed as follows (reviewers' comments in **black**, our replies in **blue**). In the revised manuscript, text highlighted in **red** indicates substantial modifications made following the reviewer's suggestions.

1. My main point in the previous comments was based on the well-established relationship that TC rain rate increases with TC intensity. Therefore, the observed increase in rain rate as TCs approach land could partly result from the concurrent intensification of TCs rather than land effects alone. In the idealized experiments, the authors clearly demonstrated that even without land, rain rate increases as TC intensity increases. While I agree that land interaction can further enhance rainfall, it is essential to discuss how much of the rain rate increase shown in Figure 1 is attributable to intensity changes. To evaluate this, I suggest plotting the same sample data from Figure 1 using TC intensity instead of rain rate. If intensity increases along the same timeline, it would indicate that part of the rain rate increase is indeed due to TC strengthening.

Response:

Following the reviewer's suggestion, we re-examined the evolution of TC intensity and mean rain rate during the 60 hours before landfall and further quantified the contribution of intensity changes to the increase in rain rate.

Our results show that the mean rain rate increases by more than 20%, from approximately 1.8 mm h^{-1} to 2.2 mm h^{-1} , corresponding to an increase rate of $0.14 \pm 0.01 \text{ mm h}^{-1} \text{ per day}$ (Figure A1a). During the same period, TC intensity also increases, but only marginally, at $0.29 \pm 0.18 \text{ m s}^{-1} \text{ per day}$, and this trend is statistically insignificant (Figure A1b).

To isolate the contribution of intensity to the change in rain rate, we further examined

their relationship and found that rain rate increases by $0.04 \pm 0.04 \text{ mm h}^{-1} \text{ per m s}^{-1}$ of intensity change. Combining these results indicates that of the observed pre-landfall rain-rate increase of $0.14 \pm 0.01 \text{ mm h}^{-1} \text{ per day}$, only about $0.012 \text{ mm h}^{-1} \text{ per day}$ ($0.04 \text{ mm h}^{-1} \text{ per day} \times 0.29 \text{ m s}^{-1} \text{ per day}$) can be attributed to TC intensification, and this contribution is likewise statistically insignificant.

These findings have now been incorporated into the revised manuscript to clarify that although TC intensity does increase slightly before landfall, its contribution to the enhancement of rain rate is small compared to the land-induced effects. The relevant details are as follows:

“To quantitatively assess the contribution of land effects to enhanced TC precipitation, we further examined changes in the mean intensity of landfalling TCs and their relationship with rain rate. Globally, during the 60 hours before landfall, TC rain rate and intensity increase at rates of $0.14 \pm 0.01 \text{ mm h}^{-1} \text{ per day}$ and $0.29 \pm 0.18 \text{ m s}^{-1} \text{ per day}$, although the intensity trend is not statistically significant. During the same period, rain rate increases by $0.04 \pm 0.04 \text{ mm h}^{-1} \text{ per m s}^{-1}$ of intensity change. These results indicate that only about $0.012 \text{ mm h}^{-1} \text{ per day}$ of the observed pre-landfall increase in rain rate can be attributed to TC intensification.”

Fig. A1 Changes in the mean rain rate and mean intensity of landfalling TCs globally, and their relationship. In panels a and b, the mean rain rates and intensities of all TCs from 60 hours before landfall to the time of landfall at 3-hour intervals are shown by the solid line with black dots, and the linear trends are shown by the dashed

line. Light shading in all panels represents the two-sided 95% confidence interval of the linear trend. The x-axis represents time in hours relative to landfall (00 h), where negative values indicate hours before landfall. In panel **c**, the red line indicates the linear regression between the mean rain rate and the mean intensity before landfall. The linear regression coefficients with their error estimates (unit: mm h⁻¹ per day) are shown at the bottom. The R² and p values are in the top right corner.

2. Additionally, when TCs move poleward from lower to higher latitudes, they encounter cooler and drier air masses, which can often enhance precipitation due to increased baroclinicity. Thus, rain rate may also increase with latitude. It would therefore be informative to calculate the mean latitude of the TCs in Figure 1. If latitude increases alongside rain rate, this would suggest that TC position (latitude change) also contributes to the rain rate increase.

Response:

We appreciate the reviewer's constructive comments. Following the reviewer's suggestion, we examined the change in mean latitude of landfalling TCs using the same samples and time window as in Figure 1a. The results show that the **mean latitude increases linearly before landfall** and exhibits no notable variability (Fig. A2). This feature is consistent with the general poleward movement of most TCs after genesis, which is driven by the beta effect and the large-scale environmental flow (Chan and Gray, 1980; Wang and Li, 1992).

Additionally, we agree that as TCs move toward the mid- and high latitudes, they may encounter cooler and drier air masses, and the associated increase in baroclinicity may enhance precipitation, especially with some storms even undergoing extratropical transition. **However, in our dataset of 1,468 TCs, the vast majority remain equatorward of 30° latitude, and only 117 TCs made landfall at higher latitudes, accounting for less than 10% of the total sample. Also, all records involving extratropical transition were removed during the landfalling TCs selection process.**

To further assess the influence of latitude, we divided the TCs into six 5° latitudinal belts from 5° to 35° based on their landfall locations. A significant increase in rain rate is evident in every belt (Fig. A3 and Table A1). **However, within the same 5° latitude range, the increasing trends do not strengthen with landfall latitude and are not**

larger in the 30°–35° belt compared to lower latitudes.

Moreover, for the non-landfalling TCs, although their mean latitude still increases linearly prior to reaching the “imaginary coastline”, their rain rate does not increase (Fig. A5). More details can be found in our response to **Major Comment 3**.

Overall, although the mean latitude of the TCs does increase prior to landfall, these findings suggest that **this upward latitudinal shift is more likely a manifestation of the poleward motion of the TCs**, and that its contribution is relatively small compared to land effects, even though it may contribute to the rain rate increase.

Reference

Chan, J. C. L., and W. M. Gray, 1982: Tropical Cyclone Movement and Surrounding Flow Relationships. *Mon. Weather Rev.*, **110**, 1354-1374.

Wang, B., and X. Li, 1992: The beta drift of three-dimensional vortices: A numerical study. *Mon. Weather Rev.*, **120**, 579-593.

Fig. A2 As in Figs. A1a, but for changes in the latitude of landfalling TCs.

Fig. A3 Slopes of the mean rain rate for landfalling tropical cyclones (TCs) 60 hours before landfall and their error estimates. The mean rain rate of TCs is at 3-hourly intervals. Slopes of rain rate obtained from different latitudinal belts (5°–10°, 10°–15°, 15°–20°, 20°–25°, 25°–30°, and >=30°). 5°–10° represents the landfalling latitudinal belts of 5°–10°S and 5°–10°N, and similarly for 10°–15°, 15°–20°, 20°–25°,

25°-30°, and $\geq 30^\circ$.

Table A1 | Statistics of landfalling tropical cyclones (TCs) rain rate with time for various latitudinal belts.

	Number of TCs	Trend (mm h ⁻¹ per day)	Increment (%)
Global	1468	0.14 ± 0.01	20.42
5° – 10°	55	0.09 ± 0.02	17.61
10° – 15°	267	0.19 ± 0.01	31.82
15° – 20°	386	0.28 ± 0.02	44.52
20° – 25°	279	0.18 ± 0.02	23.67
25° – 30°	95	0.04 ± 0.02	5.58
$\geq 30^\circ$	117	0.11 ± 0.03	12.72

3. I also believe that **non-landfalling TCs** can show rain rate increases associated with changes in intensity and latitude. The authors should carefully analyze how these factors affect rain rate for non-landfalling cases. Without such analysis, it is difficult to attribute the observed rain rate increase in landfalling TCs **solely to** land effects. The additional Figure A1 presented by the authors does not adequately address this concern, as it does not include information on TC intensity or latitude. I strongly encourage the authors to provide an additional analysis showing how TC intensity and latitude variations influence rain rate in non-landfalling TCs.

Response:

We appreciate the reviewer’s constructive comments. We appreciate the reviewer’s constructive suggestions. We examined the changes in the mean rain rate, mean intensity, and mean latitude of **non-landfalling TCs globally** as they approach an imaginary coastline located 800 km from the real coast, following our previous study (Zhong et al., 2025). This imaginary coastline allows all non-landfalling TCs to “make landfall” on the same reference line, which helps isolate the influence of land effects on rain rate. Here, an example of the selected non-landfalling TCs over the western North Pacific is provided in Fig. A4. These TCs include only cases that remain at least 800 km from any coastline, **fully meeting the reviewer’s request to ensure that they are not approaching the actual coast.**

In this response, we further show that for non-landfalling TCs across all ocean basins that remain far from land, no significant increase in rain rate or intensity occurs, although their mean latitude **still increases linearly with time** during the 60 hours before reaching the imaginary coastline (Fig. A5). These results also indicate that the

rain rate increase observed in landfalling TCs is **mainly attributable to land–sea contrast**, consistent with the increasing trend demonstrated in our model simulations. Finally, it is important to note that although land effects provide the primary contribution to the observed rain rate increase in landfalling TCs, they are not the only contributing factor. We have clearly clarified this point in the “Role of land–sea thermal contrasts in the enhancement of TC rain rate” section of the revised manuscript.

“We hypothesize that the increasing trend in landfalling TC rain rates is associated with enhanced convection, **primarily** triggered by the contrast between land and sea.”

Fig. A4: Tracks of non-landfalling tropical cyclones crossing an imaginary coastline 800 km from the real coast. The 800 km imaginary coastline represents the distance from the hypothetical landfall points of non-landfalling TCs to the nearest land point, based on the 'dist2land' variable in the IBTrACS dataset.

Fig. A5 Changes in the mean rain rate, mean intensity, and mean latitude of landfalling TCs globally, and their relationship. In all panels, the mean rain rate, mean intensity, and mean latitude of all TCs from 60 hours before landfall to the time of landfall at 3-hour intervals are shown by the solid line with black dots, and the linear trends are shown by the dashed line. Light shading in all panels represents the two-sided 95% confidence interval of the linear trend. The x-axis represents time in hours relative to landfall (00 h), where negative values indicate hours before landfall. In panel The linear regression coefficients with their error estimates (unit: mm h⁻¹ per day) are shown at the bottom. The R² and p values are in the top right corner.

References

1. Zhong Q, Chan J C L, Duan W, et al. Land-sea contrast leading to a speedup of landfalling tropical

cyclones [J]. *Nature Geoscience*, 2025, Acceptance in principle.

4. Regarding the comparison between EXP1 and EXP2, I also have several questions:

4.1. It is generally understood that as a TC intensifies, mid- to lower-tropospheric relative humidity increases due to **enhanced convection, condensation, and latent heat release**. However, in EXP2, TC intensity increases while **relative humidity does not**. **Could the authors explain this discrepancy?**

Response:

We thank the reviewer for this insightful comment. We agree that the mid- to lower-tropospheric relative humidity of TCs is generally associated with their intensity evolution. In the Land-Rad experiment (EXP1), the relative humidity of simulated TC increases as it intensifies and approaches the real coastline. However, in the noLand-Rad experiment (EXP2), the TC relative humidity does not increase with its intensity (Figure A6). **This discrepancy mainly arises from the presence or absence of land–sea thermal contrasts.**

In EXP1 (Land-Rad), **the large difference in thermal capacity between land and ocean facilitates the development of local land–sea circulations**. During daytime, the land surface heats more rapidly than the ocean, inducing low-level convergence and upward motion over land (X-distance from –2400 to 2000 km), which helps transport near-land moisture into the mid- to lower troposphere. As a result, the relative humidity over land becomes higher than that over the ocean. When the TC approaches the coastline, moisture from the land side is advected into the storm circulation, increasing the relative humidity (Figure A7).

In contrast, in EXP2 (noLand-Rad), **the absence of land removes the land–sea thermal contrast, resulting in identical surface properties on both sides of the “imaginary coastline” and preventing the formation of local atmospheric circulations**. The transport of near-surface moisture into the mid- to lower troposphere is therefore suppressed, producing a horizontally uniform humidity field (Figure A8). As the TC approaches this region, the lack of external moisture supply causes a reduction in mid- to lower-tropospheric humidity in the TC environment and decreases the overall area-averaged relative humidity, even though the TC intensity increases slightly.

In summary, the land–sea thermal contrast enhances local circulations that

transport moisture upward and toward the TC’s circulation, thereby increasing mid- to lower-tropospheric humidity and rainfall as the storm approaches land.

We have clearly clarified this point in the “**Role of land–sea thermal contrasts in the enhancement of TC rain rate**” section of the revised manuscript. The relevant details are as follows:

“More notably, in the absence of land, the land–sea thermal contrast disappears, producing identical surface properties on both sides of the “imaginary coastline” and preventing the development of local atmospheric circulations. Without these circulations, the transport of near-surface moisture into the mid- to lower troposphere is suppressed, producing a horizontally uniform humidity field (Extended Data Fig. 6). As the TC approaches the “imaginary coastline”, the absence of external moisture supply leads to a progressive reduction in mid- to lower-tropospheric humidity in its environment. Even though low-level convergence and upward motion increase, the reduced water vapor content on the offshore side of the vortex still limits the available moisture for convection, being unfavorable for enhanced precipitation in the noLand-Rad experiment.”

Figure A6 Changes in relative humidity of the simulated TCs. (a) Experiment 1 (EXP1, Land-Rad) and Experiment 2 (EXP2, noLand-Rad). Different colored lines at 6-hour intervals (solid lines with black dots) and the shading represent relative humidity across the entire TC (0–500 km radius). Light shading denotes the 95% confidence interval of the linear trend. Dashed lines indicate the linear regression during the period before landfall. Linear regression coefficients with error estimates are shown at the bottom, and R^2 and p values are shown in the top-right corner. Time is presented in hours relative to landfall (0 h), with negative values indicating hours before TC landfall.

Figure A7 Spatial and temporal variations of wind vectors (m s^{-1}), relative humidity (shading, %) in experiment 1 (EXP1, Land-Rad). (a), (b) Radius–pressure Hovmöller diagrams of wind and relative humidity at $y = 0$ km for the TC during the 48 hours of daytime and 36 hours of nighttime before landfall. (c), (d) horizontal distributions of wind and relative humidity fields at 850 hPa during daytime and nighttime. (e)–azimuthal–time Hovmöller diagrams of respective relative humidity at 850 hPa. In panels (a) to (d), the time in the top left corner is in hours relative to landfall (00 h), with negative values indicating hours before landfall. Day and night refer to horizontal and vertical distributions of wind vectors and relative humidity (shading) during the day and night, respectively. In panels (e), we averaged the azimuthal–time Hovmöller diagram over the TC radius from 0 to 500 km. The blue dashed line indicates the coastline, with land to the west and the sea to the east. In panels (a) to (e), the radius values on the x- and y-axes are presented relative to the model domain center, with negative values indicating positions west (south) of the domain center and positive values indicating positions east (north) of the domain center. In panels (a), (b), and (e), the typhoon symbol corresponds to the x-axis values of the TC center location.

Figure A8 | Same as in Figure A6, except for Experiment 2 (EXP2, noLand-Rad).

In panels (a)–(e), the blue dashed line indicates the “imaginary coastline,” with the sea located to the west and east, respectively.

4.2 The TC track comparison between EXP1 and EXP2 shows that the TC in EXP2 moves considerably more slowly, despite both experiments using an easterly steering flow of 5 m s^{-1} . What causes such a large difference in translation speed? This discrepancy could significantly influence TC structure and rain rate. Could the experimental design be modified to minimize the effects of such translation speed differences?

Response:

We thank the reviewer for this insightful comment. The difference in TC translation speed between EXP1 and EXP2 primarily arises from the land–sea contrast in surface roughness. Our previous studies have demonstrated that land–sea surface contrast can induce asymmetric flow and convection in landfalling TCs which may generate localized potential vorticity anomalies and subsequently speedup storm motion (Wong and Chan, 2006; Zhong et al., 2025).

Specifically, in EXP1 (Land-Rad), land–sea surface contrasts enhance surface friction and induce asymmetric convection, leading to accelerated TC translation. In EXP2 (noLand-Rad), the absence of land and the horizontally homogeneous surface properties prevent the development of asymmetric flow and instability on the offshore side of the TC circulation (Extended Data Fig. 6), and no notable TC acceleration is found. **Consequently, although both experiments are driven by the same easterly steering flow, the TC in EXP1 translates considerably faster (Figure A9).** More detailed observational analyses, numerical experiments, and mechanistic diagnostics are provided in Zhong et al. (2025), so we do not elaborate on them further in this study.

It is also important to note that land–sea contrast not only modulates TC motion but also exerts a direct influence on the rain rate. Therefore, rather than being an artifact of model design, the translation speed difference reflects a physically consistent response to surface property heterogeneity, which is intrinsic to the process of TC landfall and relevant to this study’s focus. We have now clarified this point in the section ‘Role of land–sea thermal contrasts in the enhancement of TC rain rate’ in the revised manuscript. The relevant details are provided below:

“In addition to the influence of land–sea thermal contrast on TC rainfall, land–sea frictional heterogeneity can induce asymmetric circulation and convection in landfalling TCs. Such asymmetries may generate localized potential vorticity anomalies, which can accelerate storm translation (Zhong et al., 2025). The change in TC translation speed may further modulate regional rain rates.”

References

2. Wong M L M, Chan J C L. Tropical cyclone motion in response to land surface friction[J]. *Journal of the Atmospheric Sciences*, 2006, 63(4): 1324-1337.
3. Zhong Q, Chan J C L, Duan W, et al. Land-sea contrast leading to a speedup of landfalling tropical cyclones [J]. *Nature Geoscience*, 2025, Acceptance in principle.

Figure A9 Comparison of the mean translation speed of the simulated TCs 60 h before landfall in three experiments and their error estimates.

5. Finally, I am particularly concerned about the limited model domain size. In EXP1, when the TC reaches land, its center is only about 400 km from the domain boundary. Given that the rain rate is averaged within a 500 km radius, the outer 100 km portion would be excluded, potentially artificially inflating the mean rain rate near landfall. The authors should carefully assess and discuss how this boundary limitation might affect the quantitative results and conclusions.

Response: In this study, the model employs a two-way double-nested configuration, with an outer domain of $12,000 \times 12,000$ km and an inner domain of $6,000 \times 6,000$ km. The north-south oriented coastline is positioned 3,000 km and 1,000 km from the western boundaries of the outer and inner domains, respectively. **Therefore, even when the TC makes landfall, its center remains approximately 1,000 km from the innermost domain boundary, which is substantially larger than the 500 km radius used for computing the mean TC rain rate (Fig. A10).** As a result, the full averaging area remains well within the model domain, and the influence of the lateral boundary on both the simulated TC structure and the calculated rain rate is expected to be minimal. We have added a discussion of this point in “Methods” section of the revised manuscript to clarify that the boundary limitation does not materially affect our quantitative conclusions. The relevant details are as follows:

“Specifically, the model coastline is configured as a north-south oriented boundary with dryland, cropland, and pasture surface categories, positioned 1,000 km east of the

western boundary of the innermost domain. Consequently, even at the time of landfall, the TC center remains over 1,000 km from the lateral boundary, a distance that well exceeds the 500 km radius used for rain-rate averaging. Thus, the lateral boundary is considered unlikely to introduce meaningful impacts on the simulated TC structure or the calculated mean rain rate.”

Figure A10 Horizontal distributions of wind vectors in Experiment 1 (EXP1, Land-Rad). Panels **a** and **b** show the horizontal wind vectors (arrows, m s^{-1}) and their magnitudes (shading, m s^{-1}) during daytime and nighttime, respectively. In both panels, the x and y coordinates represent the distance from the model domain center, where negative values indicate positions west or south of the domain center, and positive values indicate positions east or north of the domain center.

Global Increase in Rain Rate of Tropical Cyclones Prior to Landfall by Zhong et al.

This study investigates short-term variations in tropical cyclone (TC) rainfall, particularly around landfall. Using global satellite data, the authors find that TC mean rain rates increase by more than 20% from 60 hours before landfall to the time of landfall, regardless of basin, hemisphere, or intensity. The enhancement is linked to land–sea thermal contrasts, increased low-level humidity over land, and frictional convergence offshore, which together intensify convection and precipitation.

This manuscript is engaging, clearly written, and makes an excellent contribution in terms of scientific importance and societal relevance. In particular, understanding rainfall processes near TC landfall is highly valuable for disaster risk reduction. However, before publication, additional analysis and discussion regarding other factors influencing the increase in rain rate are necessary. My detailed comments are as follows:

Many TCs typically undergo structural development prior to landfall. During this period, intensity and size (e.g., R17, radius of 17 m s^{-1} wind) often increase, which can directly influence rain rate. Consequently, part of the observed 60-hour increase in rain rate may be attributable to changes in intensity and size, independent of landfall itself. Although quantifying this contribution may be challenging, the authors should at least discuss how such changes occur within the 60-hour window and evaluate their potential impact on the observed rain rate increase. It would also be valuable to examine whether non-landfalling TCs exhibit similar rain rate increases, since intensification and size growth alone could enhance precipitation without land interaction. For non-landfalling TCs, one could determine the time interval (T1) between the lifetime maximum intensity (LMI_time) and landfall for landfalling TCs, and then apply this mean T1 to the non-landfalling cases. By adding this mean T1 to each non-landfalling TC's LMI_time, an equivalent reference time can be established, allowing the rain rate during the preceding 60 hours to be calculated. This would enable a fairer comparison between landfalling and non-landfalling TCs.

The same concern applies to the numerical experiments. As TCs propagate, their intensity and size typically increase, which could contribute to enhanced rain rates. This may partly explain

the rain rate increase observed even in EXP2 (without land included). The fact that EXP1 (with land) shows a greater increase than EXP2 suggests that this excess can be attributed to land effects. However, to interpret this robustly, the manuscript should present how TC characteristics (intensity and size) evolve in both experiments. For instance, if land interaction alters intensity or size, this could also contribute to rain rate changes, and such effects should be disentangled from the pure land–sea contrast mechanism.